# Genome-Wide Identification, Evolution and Expression Analysis of *GRAS* Transcription Factor Gene Family Under Viral Stress in *Nicotiana benthamiana*

**DOI:** 10.3390/plants14152295

**Published:** 2025-07-25

**Authors:** Keyan Yao, Shuhao Cui, Songbai Zhang, Hao Cao, Long He, Jie Chen

**Affiliations:** 1Zhejiang Green Pesticide 2011 Collaborative Innovation Center, Zhejiang Agriculture and Forestry University, Hangzhou 311300, China; 15905709765@163.com (K.Y.); csh1753688409@163.com (S.C.); ch990805hao@163.com (H.C.); 2Key Laboratory of Pest Management of Horticultural Crop of Hunan Province, Hunan Academy of Agricultural Science, Changsha 410125, China; zsongb@hunaas.cn

**Keywords:** GRAS transcription factor, *Nicotiana benthamiana*, transcriptome, viral stress

## Abstract

The *GRAS* gene family not only performs a variety of regulatory functions in plant growth and development but also plays a key role in the defense mechanisms of plants in response to environmental stresses. Although *GRASs* have been identified in many species, research on them in *Nicotiana benthamiana* remains relatively limited until now. In this study, we comprehensively analyzed the *GRAS* gene family in *N. benthamiana* plants. Phylogenetic analysis displayed that all identified *NbGRASs* were classified into eight different subfamilies. Gene duplication analysis revealed that segmental duplication was the main driving force for the expansion of the *NbGRAS* gene family, with a total of 40 segmental duplication pairs identified. *NbGRASs* were unevenly distributed across the 19 chromosomes. Additionally, both gene families exhibited a relatively weak codon usage bias, a pattern shaped by mutational and selective pressures. Expression analysis showed that *NbGRASs* had tissue-specific expression patterns, with relatively high expression levels being observed in leaves and roots. The expression of *NbGRASs* was significantly changed under *tomato yellow leaf curl virus* or *bamboo mosaic virus* infection, suggesting that these *NbGRASs* can be involved in the plant’s antiviral response. These findings provide new perspectives for in-depth understanding of the evolution and functions of the *GRAS* gene family in *N. benthamiana*.

## 1. Introduction

Transcription factors (TFs) are of vital importance in controlling gene expression across diverse organisms. They participate in processes such as cellular signaling, the cell cycle, developmental stages, and responses to various stress in plants [1]. The *GRAS* family is one of the most important families of transcription factors. The designation of *GRAS* stems from the three initial members that were identified, specifically GAI (Gibberellin-Acid Insensitive), RGA (Repressor of GA1), and SCR (Scarecrow) [2,3]. GRAS proteins usually contain 360–850 amino acids [4]. The N-terminus of the *GRAS* family is highly diverse in length and sequence, while the C-terminus is highly homologous, with conserved motifs: LHRI, VHIID, LHRII, PFYRE, and SAW [5,6,7]. The VHIID motif of the C-terminal region is a core structure responsible for protein–protein interaction, which interacts with LHR I and LHR II, forming the LHR I-VHIID-LHR II complex. The variability in the N-terminal sequences of GRAS proteins, which may determine their specificity, is highly pronounced. Moreover, within the DELLA subfamily, these sequences notably include two conserved N-terminal domains: the DELLA and TVHYNP domains [7]. In early studies, the *GRASs* were divided into eight subfamilies based on *Arabidopsis thaliana* and rice, namely DELLA (the DELLA motif containing protein), HAM (hairy meristem), LAS (lateral suppressor), PAT1 (phytochrome A signal transduction 1), SHR (short root), SCR (scarecrow), SCL3 (SCR-like 3), and LISCL (lilium longiflorum SCR-like) [6].

Previous studies have revealed that GRAS transcription factors play crucial roles not only in plant growth and development but also in signal transduction, along with responses to both biotic and abiotic stresses [8,9,10]. Zhang et al. observed that overexpression of a GRAS TF, *HcSCL3* could not only promote plant growth but also induce salt stress adaptation in *Arabidopsis* [11]. The *GRAS* family member DELLA protein can regulate jasmonic acid signaling to regulate plant growth and development through the GA signaling pathway, thereby enhancing the viability of plants under stress conditions [12]. The overexpression of the VaPAT1 gene derived from Amur grape (*Vitis amurensis*) in transgenic *Arabidopsis* results in an augmentation of the plant’s resilience to abiotic stress [13]. Studies have shown that in petunia, the HAM gene encodes a putative transcription factor of the *GRAS* family. This gene not only regulates the development of lateral organ primordia and stem vascular tissue, but also controls the maintenance of the shoot apical meristem [14]. The novel *GRAS* family member *LlSCL* gene in lily is prominently expressed during meiosis and regulates related genes during microsporogenesis through the transcriptional activation activity of its amino terminus [15]. In transgenic rice, overexpressing the *OsGRAS23* could promote plant growth and bolster the rice’s tolerance to drought and oxidative stress through inducing the expression of several stress-responsive genes [16].

As a crucial model plant for exploring host–pathogen interactions, *Nicotiana benthamiana* hosts extensive application in molecular biology research and biotechnology, notably in the fields of synthetic biology and the gene editing industry [17,18,19,20]. Currently, the *GRAS* family has been identified in a variety of species, including *Oryza sativa* [5], *Arabidopsis* [21], *Zea mays* [22], Chinese cabbage [23], *Solanum lycopersicum* [24], and *Castor bean* [25]. Despite ongoing research, the *GRAS* gene family in *N. benthamiana* is underexplored. Luckily, several updated chromosome-level reference genomes of *N. benthamiana* (2n = 19) have been released, providing a solid basis for investigating the *GRAS* gene families within its genome [26,27].

A thorough genome-wide analysis of the *GRAS* gene family in *N. benthamiana* was performed using the recently available genome data [26,27]. A total of 83 *NbGRAS* members were identified in this study. Furthermore, a comprehensive analysis was conducted on the gene structure, phylogenetic relationships, motif composition, codon usage bias, and synteny of the *GRAS* gene family in *N. benthamiana* plants. The expression patterns of *NbGRAS* genes in response to viral infection were also examined. Our research will not only facilitate further investigation into *GRAS* transcription factor genes but also aid in the genetic enhancement of plant resistance.

## 2. Results

### 2.1. Identification and Characterization of the GRAS Gene Family in Nicotiana benthamiana

In order to identify all potential *GRAS* genes, we used Tbtools to perform a genome-wide search in the *Nicotiana benthamiana* genome to identify all members of the *GRAS* family [28,29]. Then, the functional annotation of all candidate genes was further confirmed through analysis with the protein family database (Pfam) and the NCBI Batch CDD tool. A total of 83 *GRAS* genes were identified in the genome of *N. benthamiana*. To gain a clearer understanding of the characteristics of *NbGRAS* genes, we analyzed their basic information, including amino acid lengths, molecular weights (MWs), theoretical isoelectric points (PIs), numbers of introns and exons and subcellular locations. The number of amino acids in *NbGRASs* ranged from 188 (*NbGRAS73*) to 874 (*NbGRAS70*), with a molecular weight range of 20.8 kDa to 99.4 kDa. The isoelectric point (pI) values of *NbGRASs* ranged from 4.61 to 9.24. The number of introns and exons varied from 0 to 3 and from 1 to 3, respectively (Appendix A). Moreover, subcellular localization analysis indicated that all identified *NbGRASs* were localized in the nucleus (Appendix A).

In order to systematically analyze the evolutionary relationships between the *GRAS* gene family of *N. benthamiana* and other representative plant species, we collected GRAS protein sequences including 23 *AtGRASs*, 60 *OsGRASs*, and 54 *SlGRASs* (Figure 1). Utilizing the IQ-TREE 2.2.5 software (IQ-TREE 2.2.5, Canberra, Australia), we constructed a maximum likelihood (ML) phylogenetic tree employing the JTT + F + R10 model. Based on the results of the phylogenetic tree, the 83 *NbGRASs* can be categorized into eight subfamilies: DELLA, SCL3, SCR, LAS, HAM, LISCL, SHR, and PAT1. Among these, the DELLA and PAT1 subfamilies have the highest number of members, with 19 and 15 members, respectively, while the LAS and SCL3 subfamilies have the fewest, with 3 and 4 members, respectively (Figure 1), which is consistent with several previous reports [23,24,28,29].

### 2.2. Analysis of Conserved Motifs and Gene Structures of the NbGRASs

In order to determine the differences in gene structure and provide a basis for the evolution of *NbGRAS* family structural diversity, the intron and exon structures of 83 *NbGRASs* were compared and visualized. The number of introns of *NbGRASs* varies from 0 to 3 (Figure 2C). Subsequently, 10 conserved motifs were identified in *NbGRASs* by MEME kit. The number of motifs for *NbGRASs* varied from 2 to 10. Five conserved motifs, including motif 1, motif 3, motif 6, motif 7 and motif 10, were highly conserved in the *NbGRAS* family, except for *NbGRAS73* (Figure 2B). Moreover, *NbGRASs* within the same subfamily exhibit relatively conserved motifs and gene structures, suggesting potential functional consistency among these members.

### 2.3. Chromosomal Distribution and Gene Duplication of the NbGRAS Family

The results of chromosome (chr) mapping showed that 83 *NbGRAS* genes were unevenly distributed on 19 chromosomes (Appendix A). Chr14 showed the highest density of *NbGRAS* genes, including nine members (10.84% of the total). The remaining *NbGRAS* genes were mostly distributed across Chr04, Chr06, Chr07, Chr08, Chr09, Chr13, Chr17 and Chr18, while the number of genes on Chr01, Chr02, Chr03, Chr05, Chr10, Chr12, Chr15, Chr16 and Chr19 ranged from only two to five. Collinearity analysis showed distinct patterns of gene duplication events in the *NbGRASs*. A set of 40 segmental duplication pairs, together with 13 tandem repeat pairs, were identified for *NbGRASs* (Figure 3). Evolutionary analysis of the *NbGRAS* gene pairs verified that all Ka/Ks ratios of them were significantly less than 1, displaying strong purifying selection pressure acting on these duplicated genes of *NbGRASs* throughout their evolutionary history.

### 2.4. Synteny Analysis of GRAS Genes

In order to explore the evolutionary relationships of *GRAS* genes in *Nicotiana benthamiana*, the collinearity characteristics of *GRAS* gene pairs in the genome of *N. benthamiana*, monocot *Oryza sativa*, dicot *Arabidopsis thaliana* and *Solanum lycopersicum* were analyzed by using the multicollinearity scanning toolkit (MCScanX, Athens, Greece) [30]. The results showed that 44 *GRAS* gene homologous pairs were identified between *A. thaliana* and *N. benthamiana* (Nb-At) (Figure 4A), 75 GRAS gene homologous pairs were found between tomato and *N. benthamiana* (Nb-Sl) (Figure 4B), while only 15 GRAS gene homologous pairs were found between *O. sativa* and *N. benthamiana* (Nb-Os) (Figure 4C). The results showed that the number of homologous *GRAS* gene pairs between *N. benthamiana* and *S. lycopersicum* is greater than that between *N. benthamiana* and *Arabidopsis* or *O. sativa*, which might be because both *N. benthamiana* and tomato belong to the *Solanaceae* family.

### 2.5. Prediction of Cis-Acting Elements in the NbGRASs Promoters

Specific cis-acting regulatory elements play a crucial role in the fine-tuning of gene expression by binding to their corresponding transcription factors [31,32]. A wealth of research has shown that these elements are involved in the processes of plant responses to various stress conditions [33,34,35]. We screened the 2000 bp promoter sequences of the *NbGRAS* genes utilizing the PlantCARE database [36]. A total of 10,580 cis-acting elements were isolated and characterized. We classified and counted most of the identified cis-acting elements based on their functions, which include phytohormones, environmental factors, stress, as well as growth and development. As shown in Figure 5B,C, we identified a total of 2842 cis-acting elements related to growth and development and 322 elements related to phytohormone responses within the *NbGRAS* gene family. These observations indicated that the *NbGRAS* family held an important position in regulating the response to various stress conditions and environmental adaptability [37,38].

### 2.6. Analysis of Codon Usage Patterns for GRAS Genes Across Different Species

In the genetic code system of organisms, each codon typically corresponds to a specific amino acid [10]. However, many amino acids can be encoded by multiple different codons, which are referred to as synonymous codons. Notably, the usage of codons in different genes or species often exhibits a preference for certain specific codons, and this preferential phenomenon in the use of synonymous codons is termed codon usage bias [39,40]. This may be of great significance for understanding the genomic evolution mechanism among related species. Through the analysis of codon usage bias (CUB) of *GRAS* genes in five species including *Oryza sativa*, *Arabidopsis thaliana*, *Nicotiana benthamiana*, *Solanum lycopersicum* and *Glycine max*, it was found that the average GC content of *GRAS* genes in these species was between 0.421 and 0.634, and the GC3s (G or C base content at the third position of synonymous codon) ranged from 0.356 to 0.769. The results showed that the average effective number of codons (ENC) of GRAS genes in monocots (*O. sativa*) was significantly lower than that in dicots (*G. max*, *A. thaliana*, *N. benthamiana* and *S. lycopersicum*) (Table 1), indicating that monocots exhibit stronger codon usage bias.

Analysis of relative synonymous codon usage (RSCU) has revealed the characteristics of codon usage bias (CUB) [41]. We observed that the RSCU patterns of *GRAS* genes in *S. lycopersicum* and *N. benthamiana* were extremely similar, which may be related to the close genetic relationship between these two plants. The similarity in RSCU patterns of *GRAS* genes between *G. max* and *O. sativa* is lower than that between *S. lycopersicum* and *N. benthamiana*. It is worth noting that as a monocotyledonous plant, the RSCU pattern of *O. sativa* is significantly different from that of *N. benthamiana*. In five species—*A. thaliana*, *N. benthamiana*, *O. sativa*, *S. lycopersicum*, and *G. max*—the majority of species do not have a high usage proportion of G and A at the third codon position, nor do they exhibit extreme preference for G or A. Instead, they tend to use two bases, C and T, to form the third codon position. In *N. benthamiana* (r^2^ = 0.02369, *p* < 0.01) (Figure 6A), *S. lycopersicum* (r^2^ = 0.2631, *p* < 0.01) (Appendix A), and *A. thaliana* (r^2^ = 0.1570, *p* < 0.01) (Appendix A), there was a relatively weak positive correlation between GC3s and GC12 in the coding sequences of *GRAS* genes. These results indicate that in dicotyledonous plants (such as *N. benthamiana*, *S. lycopersicum*, and *A. thaliana*), the codon bias of *GRAS* genes was jointly influenced by mutational pressure and natural selection. As for *O. sativa* (r^2^ = 0.6362, *p* < 0.01) (Appendix A), there was a significant positive correlation between GC3s and GC12 in the coding sequences of *GRAS* genes, which might be closely related to its taxonomic status as a monocotyledonous plant, suggesting that monocotyledonous plants may have unique evolutionary mechanisms or selective pressures in the process of codon bias formation. Interestingly, in *Glycine max* (r^2^ = 0.2362, *p* < 0.01) (Appendix A), there is a weak negative correlation between GC3s and GC12 in the coding sequences of *GRAS* genes.

### 2.7. Expression Analysis of NbGRASs in Different Tissues

In order to explore the expression patterns of *NbGRAS* genes across different tissues and organs, we detected the expression levels of all identified *NbGRASs* in different tissues, including apices, capsules, distress leaves, flowers, leaves, roots, seedlings, stems and tissue culture, based on the publicly available RNA sequencing data [8]. Tissue expression pattern analysis showed that *NbGRASs* were expressed in all detected tissues. Especially in leaf and root tissues, the *NbGRASs* genes showed higher expression levels. Among them, there were 12 genes with higher expression levels in leaf tissues and 10 genes with higher expression levels in root tissues (Figure 7). These results indicated that the *NbGRASs* exhibited high transcriptional activity in these tissues.

### 2.8. Expression Analysis of NbGRASs Under Viral Stress

In order to elucidate the potential function of *NbGRASs* in response to viral infection, this study analyzed the expression patterns of *NbGRASs* in the leaves of *Nicotiana benthamiana* after infection with *bamboo mosaic virus* (BaMV) and *tomato yellow leaf curl virus* (TYLCV) according to the published literature [42,43].

The members of the *NbGRASs* gene family showed different expression patterns for different virus infections (Figure 8). Among them, *NbGRAS3*, *NbGRAS24*, *NbGRAS47* and *NbGRAS63* showed significant up-regulation under BaMV infection; however, *NbGRAS8*, *NbGRAS10*, *NbGRAS31*, *NbGRAS51* and *NbGRAS74* showed significant down-regulation (Figure 8). After inoculation with TYLCV, the expressions of *NbGRAS5*, *NbGRAS7*, *NbGRAS8*, *NbGRAS31*, *NbGRAS43*, *NbGRAS44*, *NbGRAS51*, *NbGRAS68* and *NbGRAS80* were significantly up-regulated, while the expressions of *NbGRAS15*, *NbGRAS41* and *NbGRAS46* were significantly down-regulated. These results suggest that the *GRAS* gene in *N. benthamiana* may play an important role in virus stress response.

### 2.9. Expression Analysis of NbGRASs Under Viral Stress

After screening multiple *NbGRAS* genes by real-time quantitative polymerase chain reaction (RT-qPCR) analysis (Figure 9), it was found that the expression levels of *NbGRAS3*, *NbGRAS24* and *NbGRAS65* were significantly up-regulated at 15 days post inoculation (dpi) inoculated with *bamboo mosaic virus* (BaMV), while the expression levels of *NbGRAS8*, *NbGRAS10* and *NbGRAS68* were significantly down-regulated (Figure 9A). Fifteen days post inoculation (dpi) inoculated *tomato yellow leaf curl virus* (TYLCV), the expression levels of *NbGRAS7*, *NbGRAS51* and *NbGRAS44* were significantly increased, while the expression levels of *NbGRAS15*, *NbGRAS41* and *NbGRAS46* were significantly decreased (Figure 9B). The results showed that different *NbGRAS* family members showed different expression regulation patterns in response to BaMV and TYLCV infection.

## 3. Discussion

Functioning as evolutionarily conserved master regulators, GRAS transcription factors not only orchestrate plant growth and development but also mediate light signal perception, hormone signaling transduction, and stress-responsive adaptation to biotic and abiotic stresses. [1,44,45,46]. Significant progress has been made in the *GRAS* gene family for a series of plants with the rapid development of whole-genome sequencing technology.

Extensive research has demonstrated that the *GRAS* family is widely present in various plant species, including *Oryza sativa* [5], *Arabidopsis thaliana* [5], *Zea mays* [22], *cassava* [47], and Chinese cabbage [23]. In this study, we systematically identified the GRAS gene family for the first time based on the new chromosome-level genome in *Nicotiana benthamiana* [27], uncovering a total of 83 *NbGRAS* family members. Comparative analysis revealed that the number of *GRASs* in *N. benthamiana* was significantly higher than those reported in other plant species, such as *Solanum lycopersicum* [24], *Medicago truncatula* [48], and *Prunus mume* [49]. Notably, these findings were highly consistent with several previous studies [24,49], confirming that the number of *GRAS* genes was not only correlated with genome size, but also more likely determined by gene duplication events during species evolution. These discoveries provide us with novel insights for further research on the evolutionary mechanisms and functional diversification of the *GRAS* gene family. Phylogenetic analysis revealed that the all the identified *NbGRASs* could be classified into eight distinct subfamilies—DELLA, SCL3, SCR, LAS, HAM, LISCL, SHR, and PAT1 (Figure 1)—which was consistent with previously reported evolutionary characteristics of *GRAS* families in various plant species [23,24,48,49]. Notably, *GRAS* genes have been found in the Gv6 and OS19 subfamilies in tomato in previous studies, but not in *N. benthamiana* [24,50]. The phylogenetic tree demonstrated that most *NbGRAS* clustered within the same evolutionary clades as their orthologs from the model plant *Arabidopsis* or the economically important crop *Solanum lycopersicum* (tomato), indicating their closer phylogenetic relationships [5,24,51]. Interestingly, as members of the *Solanaceae* family, *N. benthamiana* and tomato *GRAS* genes exhibited remarkable evolutionary conservation, a phenomenon likely attributable to their close phylogenetic relationship and shared speciation events.

Gene amplification is one of the key factors promoting genome evolution, and it is also an important source of new functional genes [52]. Specifically, as two main ways of gene amplification, fragment and tandem replication are very common in the evolution of organisms. These two replication mechanisms not only enrich the diversity of genes, but also provide a genetic basis for organisms to adapt to environmental changes [53]. Studies have demonstrated that gene duplication events have contributed to the expansion of the *GRAS* gene family in plants such as rice, tomato, and *Arabidopsis* [5,24]. Through chromosomal localization analysis, we identified 13 tandemly duplicated *NbGRAS* gene pairs and 40 segmentally duplicated pairs in *N. benthamiana*. These findings highlight the pivotal role of both segmental and tandem duplication events in the expansion of *NbGRAS* genes. Comparative genomic analysis revealed that 75 orthologous gene pairs were observed between tomato and *N. benthamiana*, significantly exceeding homologous pairs in other species (Figure 4). This divergence likely stems from their shared phylogenetic position within the *Solanaceae* family. The limited number of orthologous gene pairs between *N. benthamiana* (dicot) and *O. sativa* (monocot) likely reflects their deep phylogenetic divergence within angiosperms, consistent with fundamental distinctions in monocot–dicot genome evolution.

Promoter analyses showed that the promoter region of the *NbGRASs* was tremendous in a large number of cis-acting elements related to growth and hormone response (Figure 5). These findings were of great significance as a number of studies have shown that plant hormones play a central role in regulating growth and development [54,55,56,57]. Specifically, hormones regulate plant growth and development by inducing or inhibiting the expression of related genes, and such a hormone-mediated gene expression regulation mechanism is one of the most in-depth and well-characterized plant response pathways. At present, the understanding of the interaction mechanism between the *GRASs* and plant hormones is still limited.

Codon usage bias (CUB) is widespread in plant genomes [10]. Codon usage bias (CUB) plays a key role in regulating gene expression and molecular evolution [9]. In this study, a codon usage bias (CUB) analysis was carried out on the *GRAS* gene family of *Nicotiana benthamiana*. Core parameters such as codon bias index (CBI), frequency of optimal codons (Fop), effective number of codons (ENC), GC content at the third codon position (GC3s), and overall GC content were calculated and analyzed. Comprehensive analysis based on multi-dimensional CUB indicators showed that the codon usage bias of *GRAS* genes in dicotyledonous plants (cultivated soybean *Glycine max*, *Arabidopsis thaliana*, *Nicotiana benthamiana* and *Solanum lycopersicum*) was significantly stronger than that in monocotyledonous plants (*Oryza sativa*) (Table 1). Further analysis of the relative synonymous codon usage (RSCU) of the *GRAS* gene family in five species showed that there was a conservative RSCU distribution pattern in both monocots and dicots (Figure 6C). Neutral mapping and Parity Rule 2 analysis revealed that the CUB of the *GRAS* gene family in monocots was mainly driven by natural selection, while dicots were driven by both mutation pressure and natural selection (Figure 6).

Extensive evidence verified that *GRASs* play a key role in the gibberellin signaling pathway, while only a few studies showed that a small number of *GRAS* family members may be involved in the signal transduction process of auxin and brassinosteroids signaling pathway [2,12]. In the plant hormone regulatory network, auxin shows particularly prominent versatility, and its regulatory range covers the entire life cycle from embryo formation to senescence, as well as various organ parts from root apical meristem to stem tip development [58]. The results of transcriptome analysis revealed that the expression levels of *NbGRASs* in different tissues of *N. benthamiana* showed significant differences (Figure 7).

The PAT1 subfamily showed the most significant expression level in the *GRAS* gene family, and its expression level was significantly higher than that of other subfamily members. GRAS genes were mainly expressed in stressed leaves and roots, with generally moderate to high expression levels. A number of studies have revealed that the *GRAS* gene family has conserved tissue-specific expression characteristics in different plant species. Taking cucumber as an example, Li et al. found that members of the HAM subfamily were mainly enriched in reproductive tissues such as floral organs, while subfamilies such as SCL3, HAM and PAT1 showed higher expression levels in vegetative organs such as leaves [59]. Similarly, the study by Wang and his team on soybean showed that PAT1, HAM, LISCL, SHR and SCL3 subfamilies showed obvious expression preference, especially in root and nodule tissues [60]. These findings collectively indicate that members of the *GRAS* gene family may be involved in the regulation of the development of different organs of plants through tissue-specific expression patterns. The tissue-specific expression patterns of these *NbGRAS* genes suggested that they may be involved in multiple key processes regulating plant growth and development [55,58,59].

As an important plant pathogen, plant viruses pose a serious threat to the growth and development of agricultural and forestry economic crops, often causing a significant decline in crop yield and leading to major economic losses [61,62]. Previous studies have shown that the *GRAS* signaling system plays a key role in plant growth and development regulation and biotic and abiotic stress response [5,23,24,49]. However, it is worth noting that there are relatively few studies on the function of this signaling pathway in plant–virus interaction, and its molecular mechanism has not been elucidated.

*Tomato yellow leaf curl virus* (TYLCV), a single-stranded circular DNA virus, belongs to the genus *Begomovirus* (family *Geminiviridae*). The virus is transmitted permanently through the vector of Bemisia tabaci. After infection, it can cause typical symptoms such as systemic yellowing, leaf curling, and limited growth of plants. It has been listed as a major limiting factor for the development of the global tomato industry [62,63,64]. *Bamboo mosaic virus* (BaMV) is a single-stranded positive-sense RNA virus with a genome length of about 6.4 kb, encoding five functional polypeptides. It is classified as *Alphaflexiviridae* [65,66].

Under the conditions of TYLCV and BaMV infection, we conducted a systematic study on the expression levels of the *NbGRAS* genes. The results showed that the transcription levels of several *NbGRAS* members (such as *NbGRAS3*, *NbGRAS5*, *NbGRAS15*, *NbGRAS31*, *NbGRAS47*, *NbGRAS51*, *NbGRAS65*, and *NbGRAS80*) all underwent significant changes (Figure 8). In summary, the results show that *NbGRASs* are likely to play a central role in the process of plant resistance to virus invasion and adaptation to stress environment. This important discovery not only provides a new perspective for in-depth analysis of the mechanism of *GRAS* signaling module in biotic stress response but also will strongly promote the research progress in this field.

## 4. Materials and Methods

### 4.1. Identification of the GRAS Family in Nicotiana benthamiana

The GRAS protein sequence files for *Arabidopsis thaliana* (TAIR10) and *Oryza sativa* (v7.0) were downloaded from the Phytozome 13 database (https://phytozome-next.jgi.doe.gov/, accessed on 16 October 2024). The *Nicotiana benthamiana* genome dataset was obtained from the *N. benthamiana* and tabacum Omics database (http://lifenglab.hzau.edu.cn/Nicomics/, accessed on 16 October 2024) [27]. Reference genomic data for tomatoes were sourced from the MicroTom database (https://eplant.njau.edu.cn/microTomBase/downloads.html, accessed on 16 October 2024) [67]. Additionally, the Hidden Markov Model (HMM) for the GRAS protein domain, identified as PF03514, was obtained from an online database (http://pfam.xfam.org/, accessed on 16 October 2024) [49,68]. The HMMER software (version 3.0; HMMER3, Cambridge, UK) was then employed to perform a scan analysis on the protein dataset of Nicotiana benthamiana. The *GRAS* gene family members identified were subsequently validated using the SMART protein database and the NCBI-CDD website. Members of the *GRAS* family that were present in both databases were selected and named.

The online software ExPASy and ProtParam (https://www.expasy.org/, accessed on 16 October 2024) were used to analyze the physicochemical properties of *GRAS* family members, such as molecular weight (MW), isoelectric point (pI), and amino acid count. We utilized the NCBI batch online CD-Search tool (https://www.ncbi.nlm.nih.gov/Structure/bwrpsb/bwrpsb.cgi, accessed on 16 October 2024) to calculate the number of exons, the number of introns, and the chromosomal distribution in the candidate protein sequences [69,70]. In addition, the subcellular localization of *NbGRAS* was predicted by the WoLF PSORT online program (https://wolfpsort.hgc.jp/, accessed on 16 October 2024) [71].

### 4.2. Phylogenetic Analysis of GRAS Genes

The *NbGRAS* sequences from *Nicotiana benthamiana*, *Arabidopsis thaliana*, *Oryza sativa*, and tomato were subjected to multiple sequence alignment analysis using the ClustalW software [72]. Subsequently, a maximum likelihood (ML) phylogenetic tree was constructed using the IQ-Tree 2.2.5 program (located in Canberra, Australia) [73]. Finally, the constructed phylogenetic tree was visualized and further refined using the iTOL online platform (https://itol.embl.de, accessed on 16 October 2024).

### 4.3. Chromosome Location and Collinearity Analysis

We extracted positional information of *GRAS* genes from the *Nicotiana benthamiana* and *Nicotiana tabacum* Omics database (http://lifenglab.hzau.edu.cn/Nicomics/, accessed on 16 October 2024) and utilized the MG2C v2.1 online tool (http://mg2c.iask.in/mg2c_v2.1/, accessed on 16 October 2024) to visualize the chromosomal locations of *GRAS* genes in *N. benthamiana*. The synteny of the *GRAS* gene family in *N. benthamiana*, *A. thaliana*, *Oryza sativa*, and tomato was analyzed using the MCScanX tool, and the analysis results were visualized using the TBtools software [30,69]. The proportion of nonsynonymous substitutions to synonymous substitutions (Ka/Ks) in tandem duplication sequences was ascertained by employing the Ka/Ks calculator in TBtools [74].

### 4.4. Promoter Element Analysis

To detect potential cis-acting elements in the promoter region, a 2000 base pair (bp) sequence upstream of the transcription initiation site was extracted from the *Nicotiana benthamiana* genome using TBtools. The PlantCARE web-based tool (https://bioinformatics.psb.ugent.be/webtools/plantcare/html/, accessed on 16 October 2024) was then used to analyze the cis-regulatory elements present in the promoter region of the *NbGRAS* gene. The collected data were processed using SPSS 21, followed by visualization using TBtools software [69].

### 4.5. Motif and Gene Structure Analysis

The conserved motifs within the *NbGRASs* were identified using the MEME sequence analysis tool (https://meme-suite.org/meme/, accessed on 16 October 2024) [75], with the parameter for the maximum number of motifs set to 10, while all other parameters were maintained at their default settings. Subsequently, TBtools software (TBtools, Guangzhou, China) was employed to visualize the results [74]. In addition, the advanced gene structure view function of TBtools was utilized to analyze and display the gene structures of the *NbGRAS* genes [74].

### 4.6. RNA-Seq Analysis of Expression Patterns

We obtained a set of public RNA sequencing datasets, specifically SRR6915, SRR685298, SRR696988, SRR696940, SRR697013, SRR696884, SRR696961, SRR696938, and SRR696992 [8]. To quantify the expression levels of *NbGRASs* under *tomato yellow leaf curl virus* (TYLCV) or *bamboo mosaic virus* (BaMV) infection, their gene expression datasets were obtained from several previous reports [42,43]. Sequence alignment was performed using HISAT2 (version 2.1.0) [76], and expression levels were quantified using StringTie2 (version 2.1.5) in the form of FPKM (per million alignment readings per thousand base transcripts) [77]. In addition, in order to visualize the gene expression profiles of various tissues such as terminal buds, fruits, stressed leaves, flowers, leaves, roots, seedlings, stems, and tissue culture, we used TBtools software to generate layout heat maps [69].

### 4.7. Plant Growth and Viral Stress

In an anti-free radical greenhouse, *Nicotiana benthamiana* seedlings grew at a constant temperature of 25 °C, with a photoperiod of 16 h of light (2000 lx) and 8 h of darkness. Agrobacterium strains harboring infectious clones of *tomato yellow leaf curl virus* (TYLCV) and *bamboo mosaic virus* (BaMV) were transferred onto LB agar plates containing kanamycin and rifampicin, and cultured at 28 °C for two days. Well-grown single colonies were selected and inoculated into liquid LB medium, followed by shaking incubation at 28 °C and 220 rpm for 12–16 h until the optical density (OD_600_) reached 0.6–1.0.

The bacterial cultures were transferred to EP tubes, centrifuged at 5000 rpm for eight minutes at room temperature, and the supernatants were discarded. The pellets were resuspended in infiltration buffer (1 M MgCl_2_, 10 mM MES, pH = 5.6, and 100 mM acetosyringone) to an OD_600_ of 0.05, and incubated in the dark at room temperature for 2–3 h. When the seedlings grew to the five-leaf stage, the Agrobacterium suspensions containing TYLCV and BaMV infectious clones were injected into the abaxial side of vein-free areas on healthy *N. benthamiana* leaves using a sterile syringe.

### 4.8. Codon Usage Bias Analysis

CodonW 1.4.2 (Houston, TX, USA) was used to analyze the codon usage bias of GRAS coding sequences in *Nicotiana benthamiana*, *A. thaliana*, tomato and *Oryza sativa*. Through the EMBOSS online tool, the optimal codon frequency, effective number of codons (ENC), GC content and GC3 content were calculated, and the relative synonymous codon usage frequency (RSCU) was analyzed [78].

### 4.9. RNA Isolation and Real-Time Quantitative Polymerase Chain Reaction (RT-qPCR)

Total RNA was extracted adhering to the manufacturer’s protocol employing TRIzol Reagent (Vazyme, Nanjing, China). Subsequently, single-stranded cDNA was generated through reverse transcription utilizing a First-Strand cDNA Synthesis Kit (Vazyme, Nanjing, China), in strict compliance with the provided specifications. Real-time quantitative PCR (RT-qPCR) was conducted employing an ABI QuantStudio5 Detection System (Applied Biosystems, Foster City, CA, USA) in conjunction with Hieff qPCR SYBR Green Master Mix (YEASEN, Shanghai, China). The experimental design incorporated a minimum of three biological replicates and three technical replicates per condition to ensure robust statistical analysis. The quantification of relative expression levels for the target genes was achieved via the 2^−ΔΔC(t)^ method, as detailed in reference [79]. All experiments were performed in triplicate, with data presented as mean ± standard deviation (SD). Statistical significance of differences was analyzed by Student’s *t*-test, and *p*-value < 0.05 was considered statistically significant. In each assay, the actin gene was utilized as the internal control. The primers employed in RT-qPCR are listed in Appendix A.

## 5. Conclusions

In this study, we conducted an in-depth exploration of *Nicotiana benthamiana* and successfully identified 83 *GRAS* genes, which were systematically classified into eight subfamilies. Through detailed analysis of gene structure and motifs, it was found that gene members within the same subfamily generally possess conserved motifs, and their gene structures show significant similarity. Based on the results of evolutionary analysis, segmental duplication was confirmed as the main factor driving the expansion of this gene family. In terms of tissue expression patterns, the expression of *NbGRASs* is highly tissue-specific, with high expression levels in leaves and root tissues. Further investigation into their response mechanisms under biotic stress revealed that after infection with *Tomato yellow leaf curl virus* and *Bamboo mosaic virus*, the expression levels of several *NbGRASs* changed significantly. This fully indicates that members of this gene family are widely involved in the plant’s antiviral response process. These findings not only reveal the potential functions of *NbGRASs* in plant growth and development but also provide a new perspective and theoretical basis for further elucidating their regulatory mechanisms in the process of stress adaptation.

## Figures and Tables

**Figure 1 plants-14-02295-f001:**
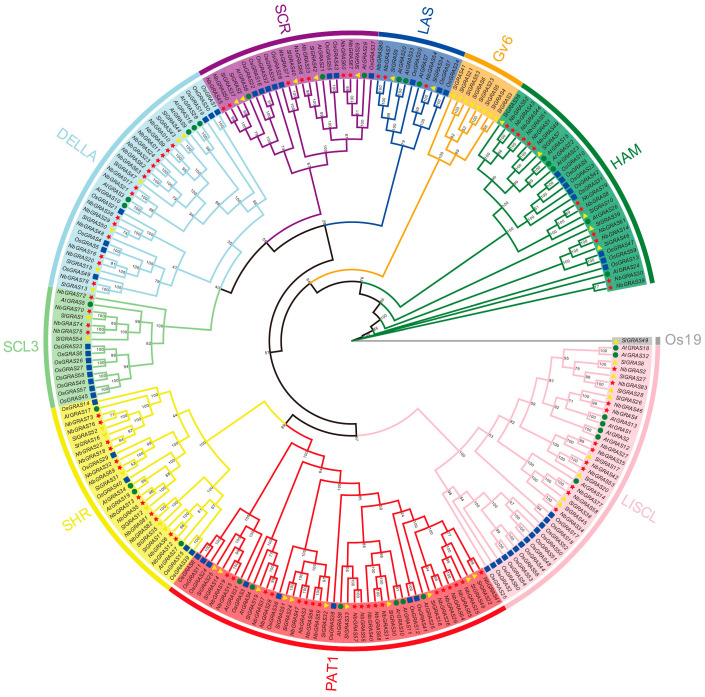
The maximum likelihood (ML) phylogenetic tree based on the GRAS amino acid sequences in *Arabidopsis thaliana*, *Nicotiana benthamiana*, *Oryza sativa*, and *Solanum lycopersicum* with IQ-TREE 2.2.5 software (IQ-TREE 2.2.5, Canberra, Australia). Green circles represent *AtGRAS*, red pentagrams represent *NbGRAS*, blue squares represent *OsGRAS*, and yellow triangles represent *SlGRAS*.

**Figure 2 plants-14-02295-f002:**
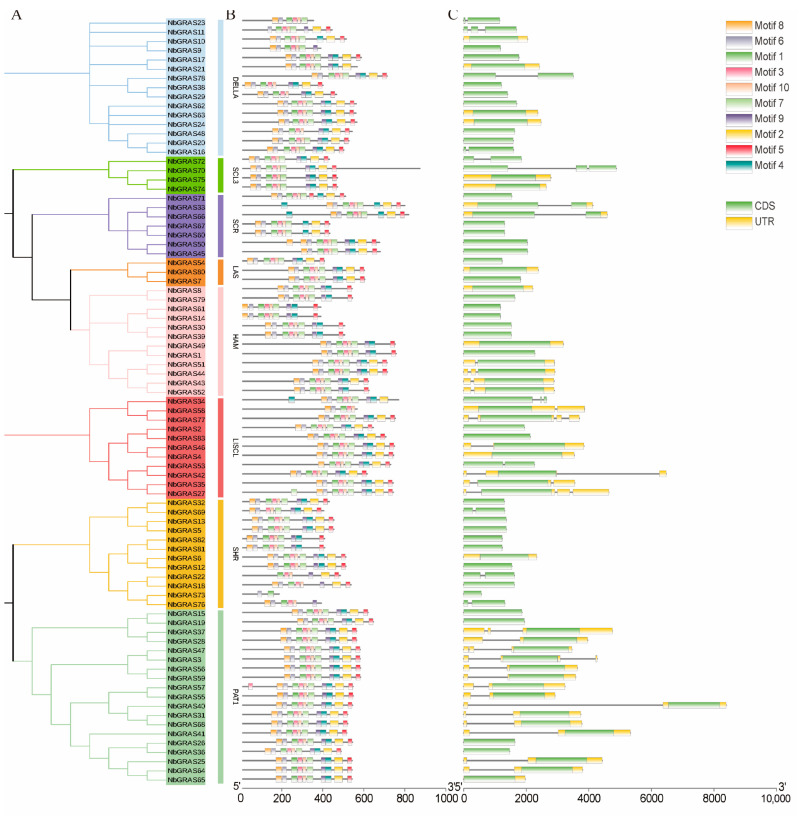
Phylogenetic relationships, motif composition, and gene structures of *NbGRASs*. (**A**) Maximum likelihood phylogenetic tree of *NbGRAS* gene family members. (**B**) Distribution of conserved motifs (numbered 1–10) among *NbGRAS*, with distinct colors representing different motifs. (**C**) Exon-intron organization of *NbGRAS* genes. Green boxes indicate coding sequences (CDS), while yellow boxes denote untranslated regions (UTRs).

**Figure 3 plants-14-02295-f003:**
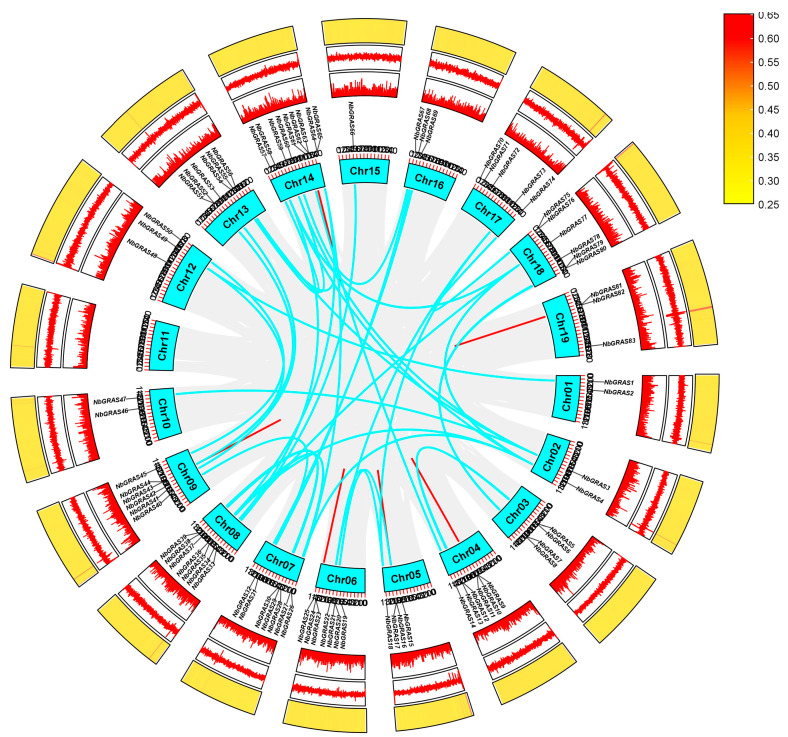
Chromosome location and gene duplication analysis of *NbGRAS* genes. Slight gray lines indicate all synteny blocks within *Nicotiana benthamiana* genome. The duplicated gene pairs are highlighted with colored lines. In the inner circle, the heat map represents chromosomal gene density; in the middle circle, it shows chromosomal GC skew; and in the outer circle, it displays chromosomal GC content.

**Figure 4 plants-14-02295-f004:**
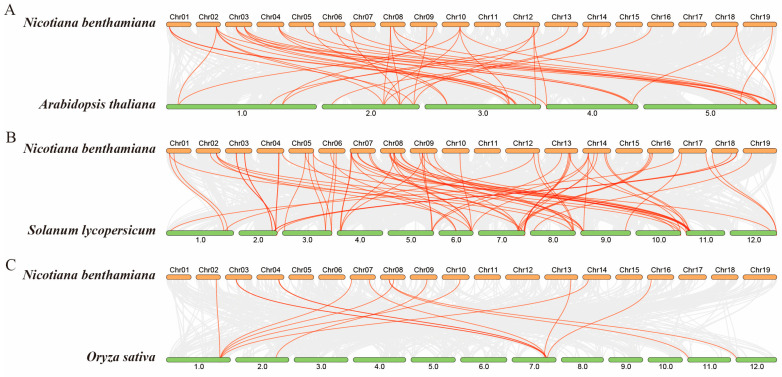
Synteny analysis of GRAS genes between *Nicotiana benthamiana* and three other species including *Arabidopsis thaliana* (**A**), *Solanum lycopersicum* (**B**), and *Oryza sativa* (**C**).

**Figure 5 plants-14-02295-f005:**
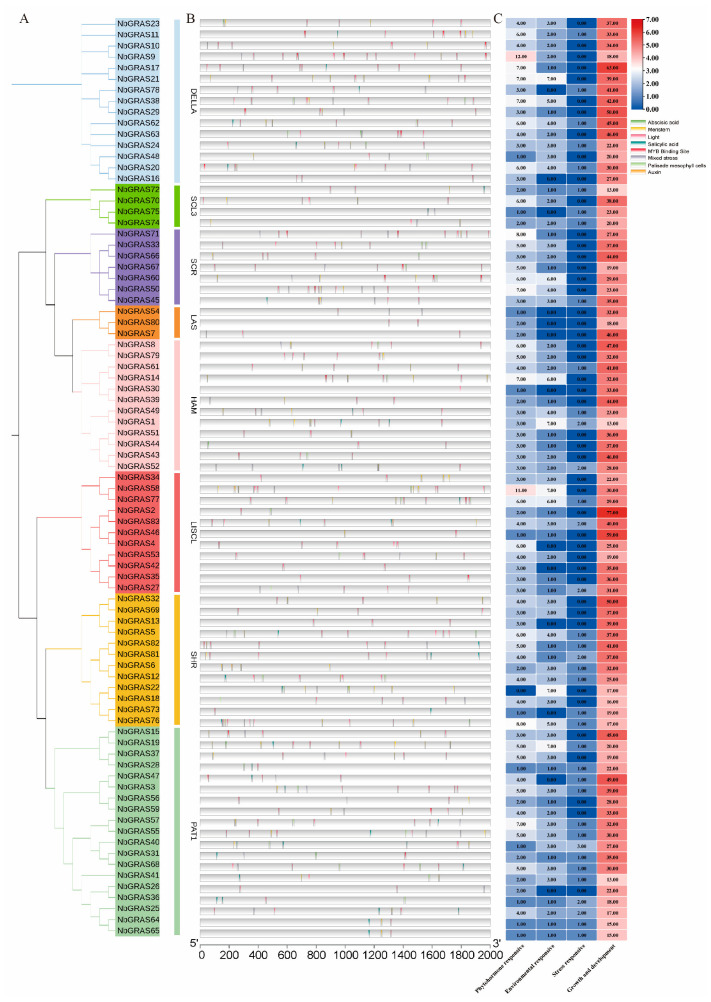
Cis-acting regulatory elements analysis of *NbGRASs*. (**A**) Phylogenetic tree of *NbGRASs*. (**B**) Summary view of cis-acting regulatory elements of *NbGRASs*. (**C**) The numbers of cis-acting regulatory elements of *NbGRASs*.

**Figure 6 plants-14-02295-f006:**
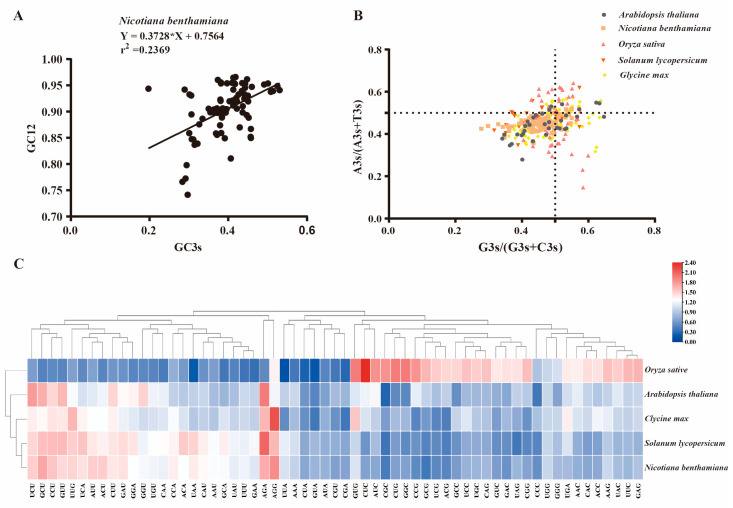
Analysis of codon usage bias: (**A**) Neutral mapping analysis was performed on all identified *NbGRAS* gene CDS sequences. (**B**) Parity rule 2 (PR2) analysis was performed on the CDS sequences of *NbGRAS* genes in five species: *A. thaliana*, *N. benthamiana*, *O. sativa*, *S. lycopersicum*, and *G. max*. (**C**) The relative synonymous codon usage (RSCU) heat map of five species (*A. thaliana*, *N. benthamiana*, *O. sativa*, *S. lycopersicum*, and *G. max*); blue to red indicates that the RSCU value of the codon is from low to high.

**Figure 7 plants-14-02295-f007:**
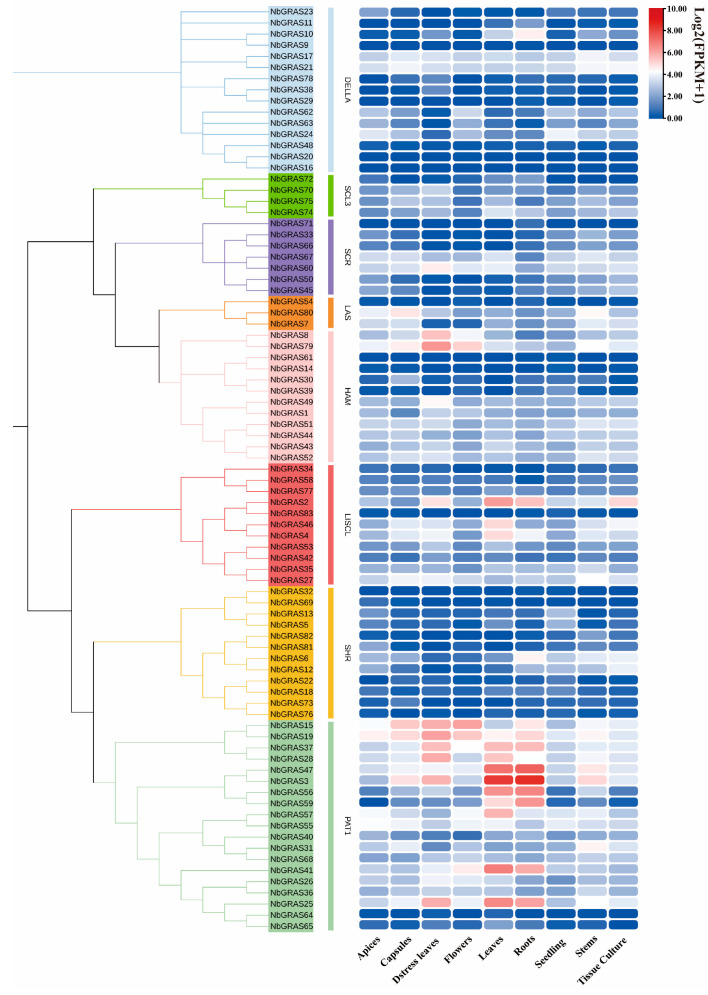
Heatmap showing the tissue-specific expression profiles of *NbGRASs*. Transcriptomic data obtained from the NCBI Sequence Read Archive (SRA) database (https://www.ncbi.nlm.nih.gov/sra/, accessed on 16 October 2024) with accession number PRJNA188486 [8].

**Figure 8 plants-14-02295-f008:**
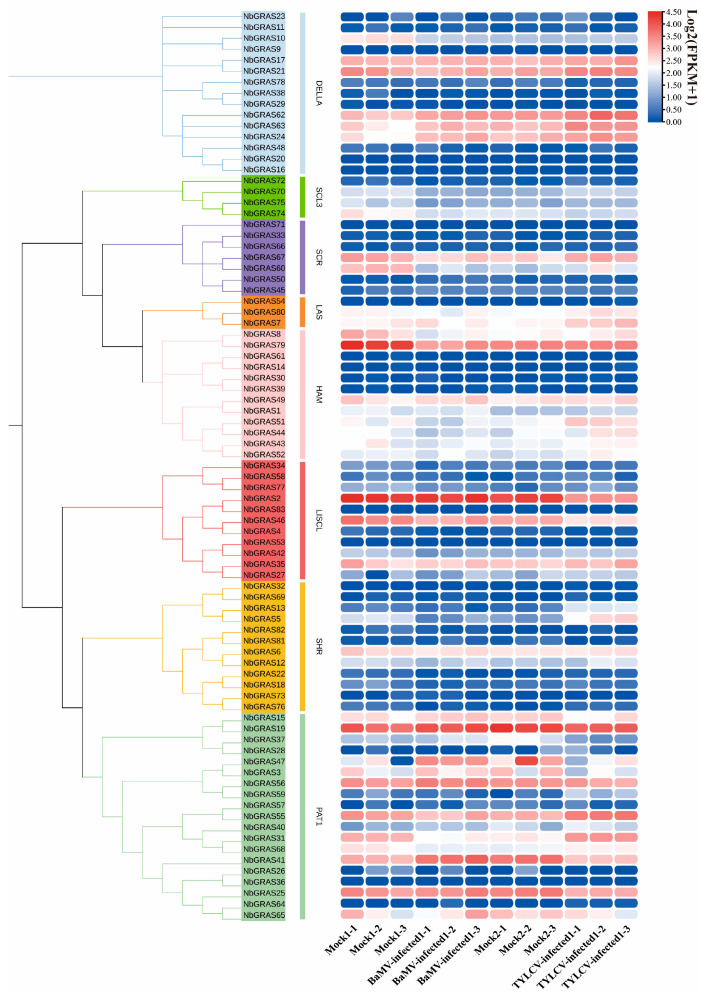
Heatmap showing expression profiles of *NbGRASs* under infection by *tomato yellow leaf curl virus* (TYLCV) and *bamboo mosaic virus* (BaMV) [42,43].

**Figure 9 plants-14-02295-f009:**
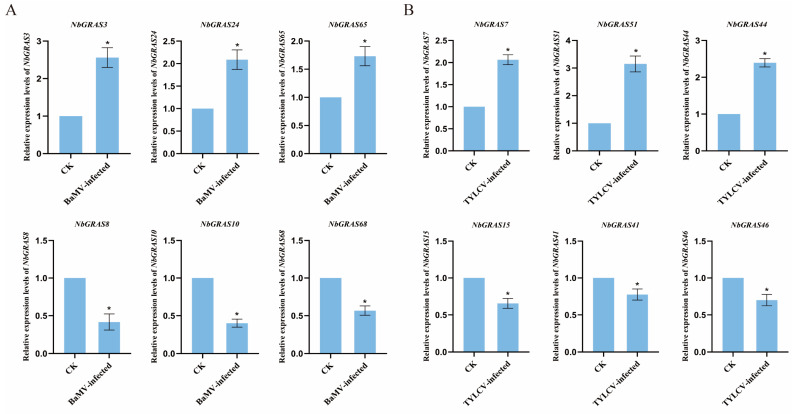
RT-qPCR results of relative expression levels of *NbGRASs*. (**A**) Relative expression levels of NbGRAS genes in BaMV-infected and control (CK). (**B**) Relative expression levels of NbGRAS genes in TYLCV-infected and control (CK). Mean ± SD values are from three biological replicates, and each replicate had three technical replicates.; *, *p* < 0.05 according to Student’s *t*-test.

**Table 1 plants-14-02295-t001:** Codon usage indicators of the *GRASs* in five different species.

Species Name	CAI	CBI	Fop	ENC	GC3s	GC Content
*Arabidopsis thaliana*	0.207	0.013	0.426	55.41	0.438	0.467
*Nicotiana benthamiana*	0.188	−0.08	0.373	53.92	0.375	0.435
*Solanum lycopersicum*	0.187	−0.082	0.372	52.98	0.356	0.421
*Oryza sativa*	0.231	0.14	0.498	45.6	0.769	0.634
*Glycine max*	0.201	−0.024	0.406	55.24	0.476	0.471

Abbreviations: CBI, codon bias index; Fop, frequency of optimal codons; ENC, effective number of codons; and GC3s, contents of G or C bases at the third position of the codons; and GC content, the contents of the G and C bases of the codons.

## Data Availability

All data generated or analyzed during this study are included in this published article and its Appendix A.

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
