# Peer review of "Genome-Wide Identification, Evolution and Expression Analysis of *GRAS* Transcription Factor Gene Family Under Viral Stress in *Nicotiana benthamiana"

_plants, 2025, doi:10.3390/plants14152295_

Round 1
Reviewer 1 Report
Comments and Suggestions for Authors
The paper describes the GRAS family transcription factors in N. benthamiana. The authors present data on identification of NbGRAS genes, analysis of conserved motifs in encoded proteins, chromosome location, prediction of functional elements in NbGRAS promoters, and tissue-specific expression of NbGRAS genes. In addition, the authors report on the analysis of the influence of virus infection on the expression of NbGRAS genes. In my opinion, this paper can be published in Plants; however, I have concerns that should be addressed before the paper is accepted for publication.
Major points
- The influence of viral infection on GRAS transcription factor expression levels (Section 2.8, Figure 8) was analyzed based on previously published data. I have three points of criticism here:
a) The paper should include a clarification of the original data used, as well as the methodology employed.
b) The authors should justify why, in this particular case, data from different independent experiments carried out under different conditions can be directly compared, as the authors do in this paper.
c) As I could understand it, the information analyzed in this part of the paper was retrieved from datasets related to the references 49-52 (line 255). However, the papers cited under references 51 and 52 contain no information on transcriptomic studies of N. benthamiana plants infected with TMV or TYLCV. This should be checked.
- The experiment on virus infection and subsequent analysis of the expression levels of selected GRAS proteins (Section 2.9, Figure 9) is not described. Some points should be clarified.
a) The source of the infectious virus clones used in the experiment must be indicated.
(b) What leaves were analyzed: agroinoculated or systemic?
(c) At what time point (dpi) was the analysis carried out, and which leaves were used for sampling?
(d) Were the levels of virus accumulation equal in the biological replicates used for analysis?
Minor points
Abstract: The first phrase can be omitted.
Line 28: infectioninfection
Line 29: These research findings – Should be These findings
Lines 73-75: This sentence should be deleted.
Line 79: a mystery – It is not a mystery, but just an unstudied subject.
Line 105: orderto
Figure 1: It should be enlarged to make gene names visible.
Line 235: Nicotiana should be N.
Section 2.9: The title of this section is identical to the title of section 2.8 and should be changed.
Figure 9: The value of relative expression is given in two different formats, 1.0 and 1, in different panels. This should be unified.
Author Response
Review1:
Comments and Suggestions for Authors
The paper describes the GRAS family transcription factors in N. benthamiana. The authors present data on identification of NbGRAS genes, analysis of conserved motifs in encoded proteins, chromosome location, prediction of functional elements in NbGRAS promoters, and tissue-specific expression of NbGRAS genes. In addition, the authors report on the analysis of the influence of virus infection on the expression of NbGRAS genes. In my opinion, this paper can be published in Plants; however, I have concerns that should be addressed before the paper is accepted for publication.
Major points
- The influence of viral infection on GRAS transcription factor expression levels (Section 2.8, Figure 8) was analyzed based on previously published data. I have three points of criticism here:
- The paper should include a clarification of the original data used, as well as the methodology employed.
Reply: Thank you for your advice. We have revised the original data used, as well as the methodology employed in new manuscript line 258 and 476-483.
- The authors should justify why, in this particular case, data from different independent experiments carried out under different conditions can be directly compared, as the authors do in this paper.
Reply: Thank you for your advice. Here, the expression levels of NbGRAS genes were compared in response to TYLCV or BaMV infection, which was then verified by RT-qPCR results in section 2.9 from line 273 to 282. These analyses together provide us a novel view to under the role GRAS transcription factor played under BaMV (harboring RNA genome) and TYLCV (harboring DNA genome)
- c) As I could understand it, the information analyzed in this part of the paper was retrieved from datasets related to the references 49-52 (line 255). However, the papers cited under references 51 and 52 contain no information on transcriptomic studies of N. benthamiana plants infected with TMV or TYLCV. This should be checked.
Reply: Thank you for your advice. We have thoroughly checked and updated the references; the relevant citations in this context should be reference 41” Importin α2 participates in RNA interference against bamboo mosaic virus accumulation in Nicotiana benthamiana via NbAGO10a-mediated small RNA clearance” and reference 42 “Transcriptional reprogramming caused by the geminivirus Tomato yellow leaf curl virus in local or systemic infections in Nicotiana benthamiana”.
- The experiment on virus infection and subsequent analysis of the expression levels of selected GRAS proteins (Section 2.9, Figure 9) is not described. Some points should be clarified.
- The source of the infectious virus clones used in the experiment must be indicated.
Reply: Thank you for your advice. The BaMV infection clone was provided by Professor Lianfeng Gu and Mingbing Zhou. The TYLCV infection clone was provided by Professor Yuzheng Mei in acknowledgments section from line 544 and 545.
(b) What leaves were analyzed: agroinoculated or systemic?
Reply: Thank you for your advice. The systemic leaves were harvested and analyzed.
(c) At what time point (dpi) was the analysis carried out, and which leaves were used for sampling?
Reply: Thank you for your advice. The systemic leaves of N. benthamiana were obtained at 15 day post inoculation (dpi) with TYLCV. The systemic leaves of N. benthamiana were obtained at 15 day post inoculation (dpi) with BaMV, which have been described more precisely from 273 to 283.
(d) Were the levels of virus accumulation equal in the biological replicates used for analysis?
Reply: We inoculated each biological replicate of N. benthamiana plant with an equal volume Agrobacterium-mediated infiltration harboring the TYLCV or BaMV infection clone , Therefore, viral accumulation levels are consistent across biological replicates.
Minor points
Abstract: The first phrase can be omitted.
Reply: Thank you for your advice. We have revised it in the Abstract Section.
Line 28: infectioninfection
Reply: Thank you for your advice. We have revised it in line 28.
Line 29: These research findings – Should be These findings
Reply: Thank you for your advice. We have revised it in line 29
Lines 73-75: This sentence should be deleted.
Reply: Thank you for your advice. We have revised it in new manuscript from line 70 to 73.
Line 79: a mystery – It is not a mystery, but just an unstudied subject.
Reply: Thank you for your advice. We have revised it in new manuscript line 80.
Line 105: orderto
Reply: Thank you for your advice. We have revised it in new manuscript line 107.
Figure 1: It should be enlarged to make gene names visible.
Reply: Thank you for your advice. Gene labels are fully resolvable in the 300-dpi JPG source file (provided separately). The reduced clarity in the manuscript draft resulted from the submission portal's automatic file compression. Full resolution will be restored in final production.
Line 235: Nicotiana should be N.
Reply: Thank you for your advice. After conducting a thorough investigation, we were unable to identify any errors in this specific instance.
Section 2.9: The title of this section is identical to the title of section 2.8 and should be changed.
Reply: Thank you for your advice. We have revised it in new manuscript
Figure 9: The value of relative expression is given in two different formats, 1.0 and 1, in different panels. This should be unified.
Reply: Thank you for your advice. We have revised it in new manuscript of figure9.
Reviewer 2 Report
Comments and Suggestions for Authors
Genome wide studies are among the popular researches today. In this study, tobacco plant was taken into consideration and important outputs were obtained. Modern analyses were used in the study and it was created with a good academic language. The references used by the authors are relevant to the subject. Some revisions are needed before publication.
The title of the study seems very complicated. Please simplify it and add the Latin name of tobacco.
Some numerical data from the study results should be added to the summary and the summary should be expanded.
There is a lack of structure in the introduction section of the study. First of all, topics such as plant characteristics and production values of the studied plant species should be mentioned.
Then, a transition from general to specific should be made. In addition, the purpose of the study can be expressed more clearly.
There are many abbreviations in the study. Please create a table of abbreviations. OR WRITE THE FULL NAME ON THE FIRST USE..
In addition to the material and method section of the study, first person singular or plural usage should be avoided (see line 422 "we")
Which type was used in the study??
I recommend that the method section be written in more detail. It is written in short sentences. Scientific methods include research that is repeatable and can be useful for later researchers.
There should be a data analysis title in the method section of the study.
When we look at the results of the study, visually good graphics are seen. However, the figures cannot be understood even at 400% magnification. In this respect, the resolution of all figures should be increased.
In addition, the discussion should be expanded and written in this section with more literature.
There are 3 different analyses in Figure 6. Can these be presented separately??
Figure s1 and Figure s2 in the additional files are sufficient in terms of resolution. Other figures should be created like this.
The conclusion section of the study should be rewritten. The findings that stand out in the study should be explained more clearly here.
In summary; there are important outputs in the study. But revisions are needed. I think it can be published after these are done.
Author Response
Review2:
Genome wide studies are among the popular researches today. In this study, tobacco plant was taken into consideration and important outputs were obtained. Modern analyses were used in the study and it was created with a good academic language. The references used by the authors are relevant to the subject. Some revisions are needed before publication.
The title of the study seems very complicated. Please simplify it and add the Latin name of tobacco.
Reply: Thank you for your advice. We have revised it in new manuscript of title.
Some numerical data from the study results should be added to the summary and the summary should be expanded.
Reply: Thank you for your advice. We have revised it in new manuscript of abstract.
There is a lack of structure in the introduction section of the study. First of all, topics such as plant characteristics and production values of the studied plant species should be mentioned.
Reply: Thank you for your advice. We have revised it in new manuscript.
Then, a transition from general to specific should be made. In addition, the purpose of the study can be expressed more clearly.
Reply: Thank you for your advice. We have revised it in new manuscript.
There are many abbreviations in the study. Please create a table of abbreviations. OR WRITE THE FULL NAME ON THE FIRST USE.
Reply: Thank you for your advice. We have revised it in new manuscript from line 557 to 566.
In addition to the material and method section of the study, first person singular or plural usage should be avoided (see line 422 "we")
Reply: Thank you for your advice. We have revised it in new manuscript.
Which type was used in the study??
I recommend that the method section be written in more detail. It is written in short sentences. Scientific methods include research that is repeatable and can be useful for later researchers.
Reply: Thank you for your advice. We have revised it in new manuscript of the method section.
There should be a data analysis title in the method section of the study.
Reply: Thank you for your advice. We have revised it in new manuscript of the method section from line 521 to 524.
When we look at the results of the study, visually good graphics are seen. However, the figures cannot be understood even at 400% magnification. In this respect, the resolution of all figures should be increased.
Reply: Thank you for your advice. Gene labels are fully resolvable in the 300-dpi JPG source file (provided separately). The reduced clarity in the manuscript draft resulted from the submission portal's automatic file compression. Full resolution will be restored in final production.
In addition, the discussion should be expanded and written in this section with more literature.
Reply: Thank you for your advice. We have revised it in new manuscript of discussion section.
There are 3 different analyses in Figure 6. Can these be presented separately??
Reply: Thank you for your advice. The content in Figure 6 exclusively focuses on codon usage bias analysis, employing three different metrics to comprehensively characterize codon preference from multiple perspectives.
Figure s1 and Figure s2 in the additional files are sufficient in terms of resolution. Other figures should be created like this.
Reply: Thank you for your advice. We have revised it in new manuscript of discussion section.
The conclusion section of the study should be rewritten. The findings that stand out in the study should be explained more clearly here.
Reply: Thank you for your advice. We have revised it in new manuscript of conclusion section.
In summary; there are important outputs in the study. But revisions are needed. I think it can be published after these are done.
Reply: Thank you for your advice.
Round 2
Reviewer 2 Report
Comments and Suggestions for Authors
Ms is ok